# Influenza A Viruses in Ruddy Turnstones (*Arenaria interpres*); Connecting Wintering and Migratory Sites with an Ecological Hotspot at Delaware Bay

**DOI:** 10.3390/v12111205

**Published:** 2020-10-22

**Authors:** Rebecca Poulson, Deborah Carter, Shelley Beville, Lawrence Niles, Amanda Dey, Clive Minton, Pamela McKenzie, Scott Krauss, Richard Webby, Robert Webster, David E. Stallknecht

**Affiliations:** 1Southeastern Cooperative Wildlife Disease Study, Department of Population Health, Wildlife Health Building, College of Veterinary Medicine, University of Georgia, 589 D.W. Brooks Drive, Athens, GA 30602, USA; dlcarter@uga.edu (D.C.); dstall@uga.edu (D.E.S.); 2St. Johns River Water Management District, Jacksonville Service Center, 7775 Baymeadows Way, Jacksonville, FL 32256, USA; sbevillecoj@gmail.com; 3Conserve Wildlife Foundation of New Jersey, P.O. Box 420, Trenton, NJ 08609, USA; larry.niles@gmail.com; 4Endangered and Nongame Species Program, New Jersey Division of Fish and Wildlife, 8747 Ferry Road, Millville, NJ 08332, USA; Amanda.Dey@dep.nj.gov; 5Victorian Wader Study Group, 165 Dalgetty Rd., Beaumaris, VIC 3193, Australia; 6Department of Infectious Diseases, St. Jude Children’s Research Hospital, Memphis, TN 38105, USA; Pamela.McKenzie@STJUDE.ORG (P.M.); krausplatz@earthlink.net (S.K.); Richard.Webby@STJUDE.ORG (R.W.); Robert.Webster@STJUDE.ORG (R.W.)

**Keywords:** *Arenaria interpres*, avian influenza, Delaware Bay, migration, ruddy turnstone, shorebird, surveillance

## Abstract

Each May for over three decades, avian influenza A viruses (IAVs) have been isolated from shorebirds and gulls (order Charadriiformes) at Delaware Bay (DE Bay), USA, which is a critical stopover site for shorebirds on their spring migration to arctic breeding grounds. At DE Bay, most isolates have been recovered from ruddy turnstones (*Arenaria interpres*), but it is unknown if this species is involved in either the maintenance or movement of these viruses outside of this site. We collected and tested fecal samples from 2823 ruddy turnstones in Florida and Georgia in the southeastern United States during four winter/spring sample periods—2010, 2011, 2012, and 2013—and during the winters of 2014/2015 and 2015/2016. Twenty-five low pathogenicity IAVs were recovered representing five subtypes (H3N4, H3N8, H5N9, H6N1, and H12N2). Many of these subtypes matched those recovered at DE Bay during the previous year or that year’s migratory cycle, suggesting that IAVs present on these southern wintering areas represent a source of virus introduction to DE Bay via migrating ruddy turnstones. Analyses of all IAV gene segments of H5N9 and H6N1 viruses recovered from ruddy turnstones at DE Bay during May 2012 and from the southeast during the spring of 2012 revealed a high level of genetic relatedness at the nucleotide level, suggesting that migrating ruddy turnstones move IAVs from wintering grounds to the DE Bay ecosystem.

## 1. Introduction

The movement of migrating birds, at both small scales and across broad latitudinal and longitudinal gradients, can serve to disseminate avian influenza A viruses (IAVs) and other pathogens over potentially long distances [1,2,3,4,5]. Wild birds in the orders Anseriformes (ducks and geese) and Charadriiformes (shorebirds, gulls, and terns) are known to have a role in the movement and maintenance of IAVs. While ducks and gulls have been shown to be critical to the long-term maintenance of IAV, shorebirds are thought to be important in the spread of viruses over long distances but on a shorter time scale [6]. Transmission of IAVs among ducks primarily occurs via a fecal-oral route [7], but less is known about transmission of IAVs among shorebirds. Within both Anseriformes and Charadriiformes, the maintenance of IAVs is presumed to be continuous, though often at a low prevalence [8], such as is the case when these birds are on their wintering grounds.

Delaware Bay (DE Bay), USA, represents a critical stopover site where shorebirds replenish fat stores and energy reserves during long distant migratory flights from their austral wintering to Arctic breeding grounds [9]. During May, large numbers of shorebirds (family Scolopacidae) and gulls (family Laridae) annually congregate at high densities (up to 210 birds per square meter) [10,11]. For more than three decades at this location, IAVs have been consistently recovered from shorebirds and gulls with most isolates originating from ruddy turnstones (*Arenaria interpres*) [12,13,14,15]. While at DE Bay, both IAV and antibody prevalence increase over time in ruddy turnstones. These increases are associated with increases in ruddy turnstones’ body mass, indicating that the majority of birds are infected locally after arrival at DE Bay [16]. This is likely a function of the high density and co-mingling of susceptible birds feeding on readily abundant horseshoe crab eggs. Migrating ruddy turnstones amplify and maintain a diversity of IAV subtypes while present at DE Bay [12,13]. At this site, the relationship between shorebirds and IAVs is localized and short-term, with ruddy turnstone-specific onsite amplification [16]. Though IAVs have been isolated from other shorebird species at DE Bay, annual IAV prevalence in ruddy turnstones has been estimated to be 10–20 times higher than in red knots (*Calidris canutus rufa*) and sanderlings (*Calidris alba*) at this site [16]. In shorebirds, recovery of IAVs outside of DE Bay and at other times of year has been infrequent [17,18,19,20], and as such, this stopover site has been coined an ecological “hotspot” for IAVs [12].

Ruddy turnstones in the Americas winter along both coasts of North and South America, and as far south as Tierra gel Fuego [21]. Upon their southern migration during late summer/early fall, these birds become widely dispersed in small flocks that frequent rocky shorelines and mudflats [21]. Winter densities are in stark contrast to those seen at DE Bay, which are estimated to be greater than 60 birds/m^2^ [11]. These birds remain on wintering grounds in the southern hemisphere and southern North America for their first complete summer [21], and the majority of adult birds return to the same breeding and overwintering locations annually, via the same migratory routes [22]. In the past decade, surveillance efforts in Brazil targeting the fall return of shorebirds from their breeding grounds resulted in the isolation of three H11N9 viruses [17], and several IAVs have been isolated from ruddy turnstones in Peru during winter months [23,24]. However, there is a scarcity of data on IAV epidemiology and ecology in shorebirds on their wintering grounds.

Despite extensive surveillance for IAVs in DE Bay, it remains unknown how these viruses (1) become so invasive and widespread in the system in such a short window of time; (2) annually reassort into dozens of different hemagglutinin (HA) and neuraminidase (NA) combinations; and (3) how IAVs are introduced into this ecological hotspot. In addition to the annual migration of shorebirds through DE Bay every spring from many distant wintering areas, this region supports breeding colonies of gulls and is host to resident and migratory waterfowl. To determine if spring migrating ruddy turnstones serve as a source of IAVs (gene segments or intact viruses) into DE Bay, we: (1) attempted to recover IAVs from this species on their wintering grounds; (2) continued ongoing IAV surveillance efforts of shorebirds at DE Bay; and (3) assessed the relatedness of isolates from these two areas.

## 2. Materials and Methods

### 2.1. Surveillance of Arenaria interpres for IAVs—Southeastern and DE Bay Sites

Fecal samples were collected from overwintering and pre-migratory ruddy turnstones (*Arenaria interpres*) at numerous Florida and Georgia sites along the southern USA coastline during 2010–2016 (Figure 1, Table 1). Collection periods consisted of the winter (December–February) and spring (March–May) seasons in 2010, 2011, 2012, 2013, and 2014/2015 and of the winter in 2015/2016 (Table 1). At DE Bay, paired oropharyngeal and cloacal (OP/CL) swabs or fecal deposits were collected from ruddy turnstones in May 2010–2015, by the combined efforts of St. Jude Children’s Research Hospital (SJCRH) and the University of Georgia (UGA).

### 2.2. Virus Isolation and Subtyping

Virus isolation and virus identification protocols for DE Bay collections by UGA and SJCRH have been previously described [12,13]. Samples collected by UGA from wintering areas were processed using these same protocols. Briefly, samples were subjected to virus isolation in 9- to 11-day old specific pathogen-free embryonated chicken eggs (ECE) as previously described [13]. After incubation of ECE at 37 °C for 120 h, amnioallantoic fluids were tested by a hemagglutination assay using 0.5% chicken red blood cells [25]. This research was conducted under University of Georgia Animal Care and Use Committee approval (A2010 6-101). Viral RNA was extracted from all hemagglutinating samples using the QIAgen Viral RNA kit (Qiagen, Inc., Valencia, CA, USA) as per the manufacturer’s recommendations. Influenza viruses were identified by reverse-transcriptase PCR (RT-PCR) targeting the matrix gene [26], and IAVs were further characterized into hemagglutinin (HA) and neuraminidase (NA) subtypes by subtype-specific RT-PCR [27] or hemagglutination (HI) and neuraminidase inhibition (NI) assay at the United States Department of Agriculture Animal and Plant Health Inspection Service National Veterinary Services Laboratory. Samples collected by SJCRH were subjected to molecular and virologic testing as previously described [28].

### 2.3. Molecular Analyses

Influenza viral RNAs from 20 isolates (H5N3, *n* = 1; H5N9, *n* = 4; H6N1, *n* = 15) were sequenced using stepwise, overlapping RT-PCR for all eight IAV gene segments, as previously reported [29]. Briefly, cDNA fragments were amplified using the one-step RT-PCR kit (Qiagen, Inc.) and previously published primers [30,31,32,33,34,35,36]. PCR products were treated with ExoSap-IT (USB Inc., Cleveland, Ohio, USA) or gel purified and extracted using the QIAquick gel extraction kit (Qiagen, Inc.) without additional purification prior to sequencing. Cycle sequencing was performed with identical primers used for RT-PCR and BigDye Terminator version 3.1 (Applied Biosystems, Foster City, California, USA); samples were analyzed on an Applied Biosystems 3730xl automated DNA sequencer (Applied Biosystems). Sequencher version 5.1 (Gene Codes Corp., Ann Arbor, MI, USA) was used to assemble, edit, and trim nucleotide contigs. MEGA version 6.041 was used to compute pairwise distance (PWD) and phylogenetic comparisons of genes sequenced in this study with North and South American internal IAV gene sequences (2000–2015; excluding sequences derived from IAVs from birds in the order Galliformes) retrieved from the NCBI Genbank Influenza Virus Resource database [37] on 21 July 2016. Given the paucity of sequence information available, reference sequences for surface genes (HA and NA) were not restricted by date. Genetic relationships between surface IAVs and all internal gene segments sequenced as part of this study and contemporary (year 2000–2015) North and South American segments were inferred in maximum-likelihood (ML) phylogenies created with 1000 bootstrap iterations. Trimmed alignment lengths for gene segments used to generate phylogenies in nucleotides are as follows: PB2 (2275), PB1 (2215), PA (2183), HA5 (1691), HA6 (1622), NP (1438), NA1 (1395), NA9 (1409), M (928), and NS (816). For these analyses, gene segments with >99.0% nucleotide (nt) identity were considered highly similar [38]. Sequences were submitted to the National Institutes of Health (NIH), National Institute of Allergy and Infectious Diseases (NIAID), Center of Excellence for Influenza Research and Surveillance Data Processing and Coordinating Center and were subsequently submitted to GenBank under accession numbers MW055264–055366 and 055368–055424 (Appendix A).

### 2.4. Barcoding Fecal Samples

Confirmation of the avian species of origin of viruses isolated from fecal samples was carried out by targeting cytochrome oxidase I (COI) mitochondrial DNA from host feces. When available, fecal swabs from which IAVs were isolated were thawed a second time. Host DNA was extracted using the QIAamp DNA Stool mini kit (Qiagen, Inc.) as per the manufacturer’s recommendations, with the following modification: 2 uL carrier RNA was added to the lysis buffer-proteinase K-sample mixture to increase DNA yield [39]. PCR amplification of the barcode regions of the COI gene were performed on DNA extracts using universal bird primers LTyr and COI907aH2 as previously described [40]. PCR products were treated with ExoSap-IT (USB Inc.) without further purification prior to sequencing. Cycle sequencing with identical primers used for PCR and sequence alignment were performed as described for influenza viral RNA. Reference Aves COI sequences (*n* = 2473) were retrieved from BOLDSYSTEMS.ORG (accessed on 18 July 2016) [41], and MEGA version 6.041 was used to generate an ML phylogenetic tree in order to infer avian species of origin.

## 3. Results

### 3.1. Virus Isolation and Subtyping

From 2823 fecal samples collected from ruddy turnstones on southeastern United States beaches prior to the northern shorebird migration, 25 IAVs were isolated (0.9% prevalence). Subtypes of viruses recovered on the Atlantic Coast of northern Florida include H3N8 (*n* = 1) in January 2011, low pathogenicity (LP) H5N9 (*n* = 3) in March (early spring) 2012, H6N1 (*n* = 9) in May (late spring) 2012, H12N2 (*n* = 1) in January 2013, and H3N4 (*n* = 11) in December 2014. Summaries of annual overwintering/pre-migration recovery of IAVs by date of collection and location are shown in Table 1. In 2012, a shift was seen in the HA/NA subtypes recovered from northeast Florida samples, from LP H5N9 in March (early spring) to H6N1 in May (late spring/migration). By season and across years, the overall IAV prevalence in the winter was 1.0% (13/1302). In the spring, 12 IAVs were recovered from 1521 samples collected (0.8%). The prevalence was highest in the months of December and May (2.7% and 2.4%, respectively). No IAVs were isolated in February or April in any years of this study.

In six of eight instances, HA and/or NA subtypes isolated from ruddy turnstones in the southeast were also recovered from DE Bay collection sites in the May preceding or following winter collections (Table 2); however, IAVs recovered from birds on wintering grounds did not represent the predominant HA/NA combinations identified at DE Bay (Table 2).

### 3.2. Molecular Analyses

Given that two different subtype combinations (LP H5N9 and H6N1) retrieved from 2012 spring samples were also present in ruddy turnstones at DE Bay three weeks to two months later, these 19 IAVs (southeast, *n* = 12; Delaware Bay, *n* = 7) were selected for full genome sequencing. One 2012 LP H5N3 from DE Bay was also included in these analyses. IAVs from Florida and DE Bay shared ≥99.0% to 100.0% nucleotide (nt) identity for individual gene segments (PWD matrices for all genes are shown in Appendix A). IAV isolates from the southeast shared greater than 99.6% nt identity for all eight gene segments. Gene segments for isolates recovered from Florida in May (late spring/pre-migration) were very similar (greater than 99.0% nt identity) to those from IAVs isolated in March (early spring) for the PB2, PB1, PA, and NS genes when compared in terms of PWD matrices (Appendix A). Nucleoprotein (NP) and matrix genes from isolates identified in the southeast in May (late spring) were very similar to one another, but differed from those earlier in the spring, sharing 98.4% and 96.0% nt identity, respectively (Appendix A). The sequences of the respective surface genes (HA5, HA6, NA1, NA9) were well conserved across time and space, with viruses recovered from ruddy turnstones in Florida and DE Bay sharing greater than 99.2% nt identity with one another within each subtype (Appendix A). Table 3 depicts the relative phenotypes, based on the 99.0% nt identity threshold for the viruses analyzed here, as well as two reference viruses that were the best matches based on an NCBI BLAST search for multiple segment sequences generated in this study: (a) reference “contributing” virus—A/Ruddy turnstone/New Jersey/AI11-1678/2011(H7N7), isolated at DE Bay in the spring the year before this study, shares high identity at the nt level to numerous 2012 southeastern and DE Bay viruses for multiple gene segments, and (b) reference “receiving” virus—A/gull/Massachusetts/13JR00943/2013(H9N1), isolated the year after the present study with four gene segments highly similar to ones identified here.

A compressed ML phylogenetic sub-tree for the surface HA6 gene is depicted in Figure 2. HA5, NA1, NA3, and NA9 are depicted in Appendix A with North and South American reference sequences; no time restriction was imposed on reference surface genes. Sequences within a respective HA or NA subtype grouped closely together in all cases, with the exception of the HA5 of AI12-2871 and the NA1 of AI12-2375. The NA3 segment of AI12-2871 was most closely related to KJ413442 A/blue-winged teal/Texas/AI12-614/2012(H10N3).

Phylogenetic trees for internal genes (PB2, PB1, PA, NP, matrix, NS) are shown in Appendix A. For all internal gene segments, gene sequences derived from ruddy turnstones collected in both the southeast and DE Bay, form clades that are shorebird-specific. Further, at least one gene sequence derived from a southeastern ruddy turnstone IAV isolate falls within clades of DE Bay gene sequences for all eight genes.

### 3.3. Fecal Barcoding

Host DNA was successfully extracted from three fecal swabs (overwintering, *n* = 2, and DE Bay, *n* = 1) from which IAVs were isolated, and identified to be most closely related to *Arenaria interpres* through analysis of a 431-base-pair portion of the coding region of the COI mitochondrial gene.

## 4. Discussion

DE Bay is the only documented “hotspot” for IAVs in shorebirds worldwide, but the dynamics of this system—namely the expansion, amplification and reassortment of IAV subtypes—are temporally short-lived and not fully understood. This important migratory stopover site represents a location where IAVs can be reliably isolated from shorebirds during a four- to five-week window of time every year, corresponding to a time of high habitat utilization by these species. While IAV prevalence in ruddy turnstones in May at DE Bay can be as high as 18% [13], the source(s) of IAVs in this unique system remains largely unknown. Sampling of shorebirds on wintering grounds, from which birds sampled at DE Bay originate, can be challenging, as many species such as ruddy turnstones are widely dispersed at relatively low densities across wintering grounds extending from the southern United States into Central and South America. Although reports of the isolation of IAVs from this species prior to spring migration are rare, genetic analyses of several H11N9 IAVs isolated from ruddy turnstones in Brazil in November provided evidence that ruddy turnstones are infected on these wintering areas and that these IAVs were genetically related to IAVs isolated from this same species at DE Bay [17,18,19]. Genetic analyses of IAVs recovered in Peru from ruddy turnstones and other Charadriiformes across several years in both early and late winter have shown that they are of North American lineage; segments from some of these viruses group closely with those from DE Bay collected the previous spring [23,24]. Gene segments from Chilean IAV isolates collected in early winter months from Franklin’s gulls (*Leucophaeus pipixcan*; H13N2 and H13N9) and a kelp gull (*Larus dominicanus*; LP H5N9) were also genetically similar to segments from IAVs isolated from shorebirds and gulls a year or two prior; all but the NA and matrix gene of the H13N9 virus were most closely matched to DE Bay IAV gene segments [42]. A limited number of IAVs have also been isolated from shorebirds at DE Bay, outside of the month of May. Kawaoka et al. [14] reported the isolation of IAVs from fecal samples collected in June and September. Though historically rare, such isolations support the idea of (1) the movement of viruses or viral gene segments with migrating shorebirds upon their fall southward migration and/or (2) the low-level circulation of IAVs on wintering grounds.

The work presented here reports the successful isolation of IAVs (*n* = 25) from fecal deposits collected at several sites along the Atlantic coast of Florida prior to the spring migration of shorebirds, between February 2010 and December 2015. In most cases, the same HA and/or NA subtypes were recovered from shorebirds sampled at DE Bay the spring prior to or immediately after these winter collections. Across all IAVs analyzed here, but not for a single IAV, full genome sequencing of the matched HA and NA subtype combinations in 2012 indicates a high degree of genetic relatedness for all eight gene segments between both early and late winter and spring viruses, including for the highly mutable surface antigens (HA and NA). Reassortment plays an important role in the expansion and maintenance of IAV diversity at DE Bay [6,7,8,9,10,11,12,13,14,15,16,17,18,19,20,21,22,23,24,25,26,27,28,29,30,31,32,33,34,35,36,37,38,39,40,41,42,43] and the dynamics of this diversity “machine” are evident (Figure 2, Table 3), even with the small number of viruses compared here. Though we did not detect a full IAV genome from DE Bay ruddy turnstones that was homologous to those of overwintering viruses, viral gene segments detected from shorebirds on their wintering grounds are being transported to and becoming assimilated into the larger IAV gene pool at DE Bay (Table 3). Successful COI barcoding of several of the fecal samples from which IAVs were derived to the level of avian species indicates that these viruses were isolated from ruddy turnstones, a shorebird species identified as playing a critical role in IAV ecology and epidemiology [12,13,14,15].

Samples collected from ruddy turnstones on their wintering grounds differ from samples collected at DE Bay, as the former potentially include hatch-year birds. Because we were sampling feces on wintering grounds, we cannot discount the inclusion of samples from this first-year age class. Ruddy turnstones do not migrate north to Arctic breeding grounds in their first year of life [21] and the circulation of viral subtypes in this age class might not be expected to be reflected in viruses recovered in DE Bay. However, the high nucleotide similarity for all eight IAV gene segments between viruses collected in winter months and then at DE Bay suggests that the infection of post-breeding birds is occurring on the wintering grounds, with subsequent movement of viral gene segments upon northern migration. The viruses recovered on wintering grounds also do not represent the predominant subtype combination recovered at DE Bay prior to or after winter sampling (Table 2). Winter isolations of IAVs from ruddy turnstones and gulls of H11N9 in Brazil [17]; LP H5N9, H13N2, and H13N9 in Chile [41]; and H10N9 in Peru [23] are also not reflective of the predominant subtypes recovered at DE Bay in those years [13]. It is possible, especially related to those viral subtypes that were detected prior to migration, that the lower prevalence of infection at DE Bay was limited by pre-exposure to the same virus and subsequent immunity acquired on the wintering grounds. In addition, exposure and longer lasting immunity acquired at DE Bay could limit infection on subsequent migrations through DE Bay.

The potential involvement of non-breeding age classes on wintering grounds in the annual maintenance of these viruses deserves additional study. Such maintenance could be enhanced through favorable environmental conditions during winter. The daily air temperature when the LP H5N9 viruses were recovered in Florida in March 2012 was 19 °C (range 13 °C–25 °C) [44]. In laboratory models, avian IAV can persist, on average, more than 20 days in distilled water at 23 °C. At 17 °C, average IAV persistence increases to more than 30 days [45]. Given that shorebirds tend to be widely dispersed on wintering grounds, the longevity of infectious IAVs in this cool winter temperature range may serve to increase viral availability to susceptible hosts. The environmental/substrate persistence of IAVs reported previously [46,47,48,49], especially as related to the unique foraging/scavenging behavior of ruddy turnstones, may also play a role in the maintenance and transmission of viruses during the winter.

Genetic analyses of other IAVs detected in overwintering shorebirds and continued surveillance of winter populations of ruddy turnstones will allow us to better elucidate the relatedness of viruses recovered at different points along a migratory pathway, both from breeding to wintering sites and vice versa. Given that we did not detect the movement of an intact virus from southeastern sites into DE Bay, connecting other species- or location-related sources of viruses and their gene segments to this ecological hotspot also warrants further investigation. At present, it is not known if these southeastern sites are simply wintering or early spring stopover points on the path to DE Bay or quality alternative habitats for shorebird refueling during migration. The unknown connectivity between these habitats and DE Bay may also have disease transmission consequences that have not yet been adequately explored.

## Figures and Tables

**Figure 1 viruses-12-01205-f001:**
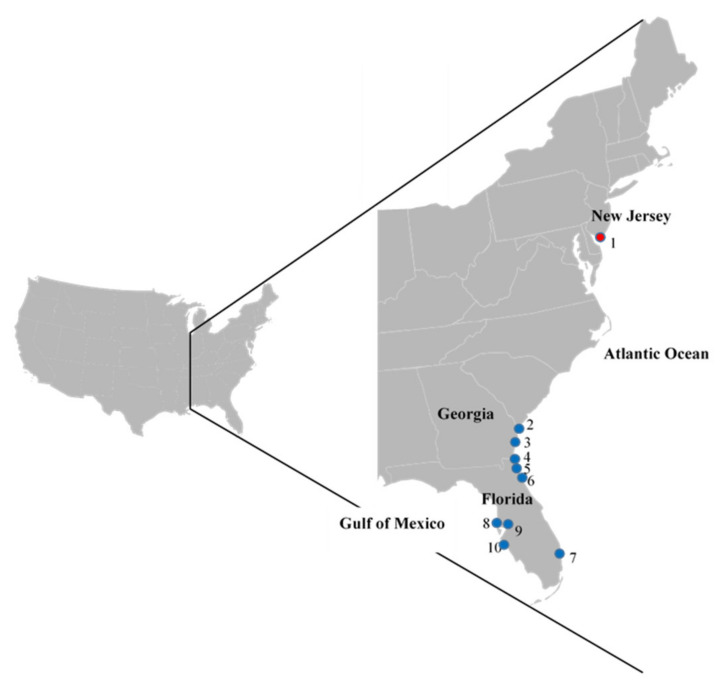
Relative location of collection sites in Florida and Georgia, USA, depicted by blue markers; the Delaware Bay spring migratory stopover site is designated with a red marker (1). States and water body names are included for orientation. Southeastern locations, by county (Co.) include: (2) Chatham Co., GA; (3) Glynn Co., GA; (4) Nassau Co., FL; (5) Duval Co., FL; (6) St. John’s Co., FL; (7) Palm Beach Co., FL; (8) Pinellas Co., FL; (9) Hillsborough Co., FL; and (10) Longboat Keys, FL. The map of the contiguous United States and the adapted inset were used under CC BY-SA 3.0 (https://creativecommons.org/licenses/by-sa/3).

**Figure 2 viruses-12-01205-f002:**
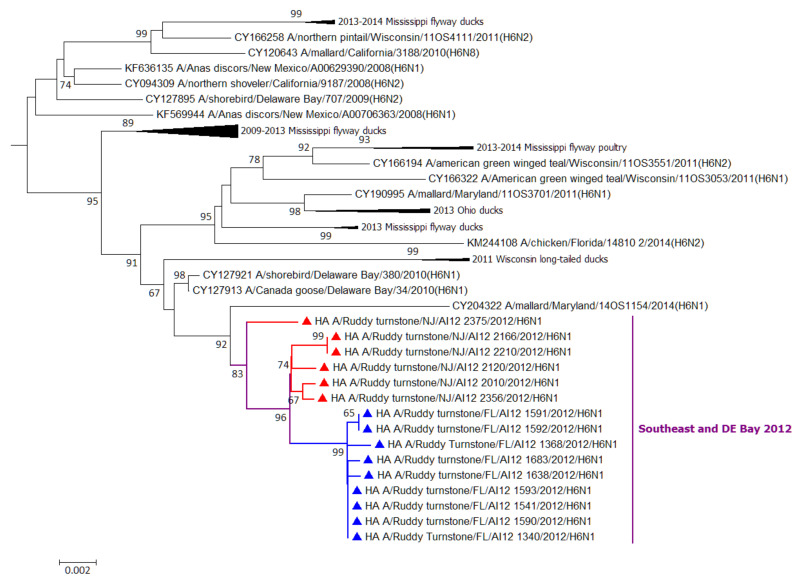
Maximum-likelihood (ML) phylogenetic sub-tree for hemagglutinin HA6 gene segments derived from influenza A viruses isolated from wild and domestic birds in North and South America (excluding Alaska), without date restriction. Nodes for HA6 segments identified in this study are colored in red (Delaware Bay) or blue (southeast) triangles. Bootstrap values lower than 65 are omitted. Branch lengths are measured in the number of nucleotide substitutions per site.

**Table 1 viruses-12-01205-t001:** Estimated influenza A virus (IAV) prevalence and subtype recovery from shorebird fecal samples collected at southeastern United States sites on the Atlantic or Gulf of Mexico Coasts, prior to shorebird migration through Delaware Bay, from 2010–2016. Bold entries indicate collection months and locations from which IAV were isolated from shorebirds.

Year	Month	Location (Figure 1)	IAV Isolated/Total Collected (% Recovery)	Subtypes (#)
2010	February	Chatham Co. GA (2)	0/8	
		Duval Co. FL (5)	0/26	
	March	Chatham Co. GA (2)	0/50	
		Pinellas Co. FL (8)	0/106	
	April	Chatham Co. GA (2)	0/71	
		Duval Co. FL (5)	0/26	
		Longboat Keys (10)	0/14	
2011	**January**	**Chatham Co. GA (2)**	**1/39 (2.6)**	**H3N8 (1)**
		Duval Co. FL (5)	**0/7**	
	February	Chatham Co. GA (2)	0/81	
		Duval Co. FL (5)	0/99	
	March	Chatham Co. GA (2)	0/118	
		Duval Co. FL (5)	0/85	
	April	Chatham Co. GA (2)	0/1	
		Duval Co. FL (5)	0/50	
2012	January	Duval Co. FL (5)	0/68	
		St. John’s Co. FL (6)	0/31	
	**March**	**Duval Co. FL (5)**	**3/64 (4.7)**	**LPAI H5N9 (3)**
		Pinellas Co. FL (8)	0/19	
	**May**	**Duval Co. FL (5)**	**9/378 (2.4)**	**H6N1 (9)**
2013	**January**	**Nassau Co. FL (4)**	**1/36 (2.8)**	**H12N2 (1)**
	March	Duval Co. FL (5)	0/164	
	April	Duval Co. FL (5)	0/1	
		Nassau Co. FL (4)	0/199	
2014	**December**	**Duval Co. FL (5)**	**7/13 (53.8)**	**H3N4 (7)**
		**Nassau Co. FL (4)**	**4/176 (2.2)**	**H3N4 (4)**
		Palm Beach Co. FL (7)	**0/19**	
2015	January	Chatham Co. GA (2)	0/69	
		Glynn Co. GA (3)	0/65	
		Nassau Co. FL (4)	0/89	
	February	Duval Co. FL (5)	0/12	
		Hillsborough Co. FL (9)	0/34	
		Nassau Co. FL (4)	0/26	
		Pinellas Co. FL (8)	0/52	
	March	Duval Co. FL (5)	0/94	
		Nassau Co. FL (4)	0/80	
		Pinellas Co. FL (8)	0/1	
	December	Duval Co. FL (5)	0/109	
		Nassau Co. FL (4)	0/92	
2016	February	Duval Co. FL (5)	0/109	
		Nassau Co. FL (4)	0/42	
		All Sites (2–10)	25/2823 (0.9)	
		Atlantic Coast (2–7)	25/2597 (1.0)	
		Gulf Coast (8–10)	0/226	

**Table 2 viruses-12-01205-t002:** Number of influenza A viruses, by subtype (number), isolated from shorebird samples collected at Delaware Bay (DB) by St. Jude Children’s Research Hospital (SJCRH, fecal samples) or at DB and southeastern U.S. sites by the University of Georgia (UGA, fecal or oropharyngeal and cloacal (OP/CL) swabs) from ruddy turnstones, by year. Collections are indicated in chronological order, from May 2010 (DB) through May 2015 (DB). HA or NA subtypes colored red were seen in a preceding or subsequent sampling effort, also designated by a red arrow along the top of the table. Cells shaded gray indicate the recovery of a matched HA and NA subtype in a preceding and/or subsequent collection.

DB 2010	Southeast 2010/2011^c^	DB 2011	Southeast 2011/2012	DB 2012	Southeast 2012/2013	DB 2013	Southeast 2013/2014	DB 2014	Southeast 2014/2015	DB 2015
H6N1 (13) ^b^		H7N3 (16) ^b^		H1N1 (38)		H10N7 (90) ^b^		H12N4 (69)		H7N3 (99) ^b^
H8N4 (4)		H9N7 (10) ^b^		H12N3 (27) ^b^	H12N2 (1)	H10N8 (33) ^b^		H13N6 (16) ^b^		H1N1 (71) ^b^
H5N2 (3)		H5N2 (9) ^b^		H12N1 (15)		H10N1 (22) ^b^		H6N2 (9) ^b^		H1N3 (21) ^b^
H6N8 (2) ^b^		H9N2 (3) ^b^		H1N8 (10) ^b^		H10N2 (20) ^b^		H11N2 (8) ^b^		H7N1 (12)
H2N3 (1)		H7N7 (2)	**H6N1(9)**	**H6N1 (8) ^b^**		H10N9 (18) ^b^		H6N4 (6) ^a^		H1N2 (2) ^a^
H2N9 (1)		H10N6 (2) ^a^		H6N4 (4) ^a^		H11N2 (17) ^b^	***No Collection***	H3N6 (5) ^b^		H1N8 (2) ^b^
H3N2 (1)		H10N9 (2)		H7N7 (4)		H11N8 (7)		H6N1 (5)		H11N2 (2) ^b^
**H3N8 (1)**	**H3N8 (1)**	H5N3 (1) ^a^		H12N8 (2)		H11N7 (5) ^b^		**H3N4 (2)**	**H3N4 (11)**	H16N3 (2) ^a^
H6N4 (1) ^a^		H5N6 (1)		H13N6(2) ^b^		H6N8 (3) ^b^		H6N8 (2) ^b^		H6N8 (1) ^a^
H13N6 (1) ^a^		H7N2 (1) ^a^		H16N6 (2) ^a^		H1N8 (1) ^a^		H1N8 (1)		
		H9N6 (1) ^a^		H1N3 (1)		H16N3 (1)		H4N4 (1)		
		H10N3 (1)		H5N3(1)				H7N3 (1)		
		**H5N9 (1)^a^**	**H5N9 (3)**	**H5N9(1)**				H12N1 (1)		
		H11N2 (1)^a^		H7N3 (1)				H16N6 (1) ^a^		
		H11N3 (1)		H9N1 (1)						
		H12N2 (1)		H10N8 (1)						

^a^ Number of isolates recovered from SJCRH fecal samples; ^b^ Number of combined isolates recovered at DB from SJCRH fecal samples and UGA fecal and/or OP/CL samples; ^c^ All southeastern viruses are from UGA fecal swabs.

**Table 3 viruses-12-01205-t003:** Viruses recovered during this study that share differing degrees of nucleotide (nt) identity within each gene segment. Viruses with the same color code within a given gene segment share >99.0% nt identity. The HA and NA of the first H5N9 (AI12-1161) and H6N1 (AI12-1541) viruses isolated are the reference sequences within those respective gene segments.

Virus	Percent Nucleotide Similarity to Reference Sequence ^3^								
	PB2	PB1	PA	NP	MA	NS	HA	NA	
A/Ruddy turnstone/NJ ^1^/AI11-1678/2011(H7N7)	*ref*	*ref*	*ref*	*ref*	*ref*	*ref*	*N/A (H7)*	*N/A (N7)*	Reference ^4^
A/Ruddy turnstone/FL ^2^/AI12-1161/2012/H5N9	92.0	99.6	99.6	91.9	96.9	95.1	*ref*	*ref*	Southeastern viruses —this study
A/Ruddy turnstone/FL/AI12-1244/2012/H5N9	92.0	99.6	99.6	91.9	96.9	95.1	99.9	100.0	
A/Ruddy turnstone/FL/AI12-1476/2012/H5N9	91.9	99.6	99.6	91.9	96.9	95.2	99.8	100.0	
A/Ruddy turnstone/FL/AI12-1541/2012/H6N1	91.9	99.3	99.4	91.7	99.1	95.2	*ref*	*ref*	
A/Ruddy turnstone/FL/AI12-1590/2012/H6N1	91.7	99.3	99.4	91.7	99.1	95.2	100.0	100.0	
A/Ruddy turnstone/FL/AI12-1591/2012/H6N1	91.9	99.2	99.4	91.7	99.1	95.2	99.9	100.0	
A/Ruddy turnstone/FL/AI12-1592/2012/H6N1	91.9	99.2	99.4	91.7	99.1	95.2	99.9	100.0	
A/Ruddy turnstone/FL/AI12-1593/2012/H6N1	91.9	99.3	99.4	91.7	99.1	95.2	100.0	100.0	
A/Ruddy turnstone/FL/AI12-1683/2012/H6N1	91.9	99.3	99.4	91.7	99.1	95.2	99.9	100.0	
A/Ruddy turnstone/FL/AI12-1638/2012/H6N1	91.9	99.3	99.4	91.7	99.1	95.2	99.9	99.9	
A/Ruddy turnstone/FL/AI12-1340/2012/H6N1	91.9	99.3	99.4	91.7	99.1	95.2	100.0	100.0	
A/Ruddy turnstone/FL/AI12-1368/2012/H6N1	92.0	99.3	99.5	93.7	99.1	95.1	99.9	99.9	
A/Ruddy turnstone/NJ/AI12-2120/2012/H6N1	99.6	99.5	87.4	91.7	99.7	98.0	99.6	99.6	DE Bay viruses —this study
A/Ruddy turnstone/NJ/AI12-2279/2012/H5N9	99.6	99.5	87.4	91.7	99.5	95.3	99.4	99.3	
A/Ruddy turnstone/NJ/AI12-2166/2012/H6N1	99.6	96.1	87.4	91.7	96.7	98.2	99.5	99.5	
A/Ruddy turnstone/NJ/AI12-2356/2012/H6N1	99.6	99.4	87.5	91.7	99.7	98.2	99.6	99.6	
A/Ruddy turnstone/NJ/AI12-2375/2012/H6N1	91.8	93.5	99.4	93.8	96.8	95.1	99.2	93.0	
A/Ruddy turnstone/NJ/AI12-2871/2012/H5N3	96.8	94.1	87.4	91.6	96.9	97.8	97.7	*N/A*	
A/Ruddy turnstone/NJ/AI12-2010/2012/H6N1	99.6	99.4	87.4	91.7	99.6	98.2	99.6	99.6	
A/Ruddy turnstone/NJ/AI12-2210/2012/H6N1	99.6	96.2	87.4	91.7	96.9	98.3	99.5	99.6	
A/gull/MA ^3^/13JR00943/2013(H9N1)	91.9	96.0	87.7	93.7	97.5	97.0	*N/A (H9)*	93.0	Reference ^5^

^1^ New Jersey (NJ); ^2^ Florida (FL); ^3^ Massachusetts (MA) ^4^ Virus A/Ruddy turnstone/NJ/AI11-1678/2011/H7N7 was used as a “contributing” reference virus for internal genes; ^5^ Virus A/gull/MA/13JR00943/2013/H9N1 was used as a “receiving” reference sequence for internal genes and NA1.

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
