# Peer review of "Influenza A Viruses in Ruddy Turnstones (*Arenaria interpres*); Connecting Wintering and Migratory Sites with an Ecological Hotspot at Delaware Bay"

_viruses, 2020, doi:10.3390/v12111205_

Round 1

Reviewer 1 Report

Summary:

The authors present results from IAV surveillance in ruddy turnstones at Delaware Bay and the Southeastern US.  Comparison of the genomes of virus isolates shows similarity between those from isolates collected at the wintering and summering grounds suggesting ruddy turnstones transport IAV during migration.  

Major Comments:

  1. Table 3. Phenotype is used incorrectly. This table is showing percent identity of nucleotide sequences.

Minor Comments:

  1. Line 56: “These increases are associated with increases in ruddy turnstone’s body mass, indicating that the majority of birds are infected after arrival.” Please clarify what is meant in this sentence; prevalence increases simultaneously with body mass?  Is weight-loss not associated with IAV infection in ruddy turnstones?
  2. Line 60: The authors state there is “ruddy turnstone-specific amplification” of IAV. Could you please clarify if the amplification of IAV occurs only in this species during migration at DE Bay?  Have other species been studied?
  3. Lines 68: Please clarify what is meant by “these birds remain on wintering grounds for their first complete summer”. Are wintering grounds in the Southern Hemisphere?  Is the term “wintering grounds” referring to the season in the Northern Hemisphere?
  4. Lines 76-77: “annually genetically diversify…” meaning reassortment?
  5. Line 184: PWD = pairwise distance?
  6. Line 328: DE Bay misspelled.

Author Response

Please see the attachment. Thank you for your helpful review!

Reviewer 2 Report

The science behind this study is good and the results are well presented.

My major concern is the discussion.  The first two paragraphs background information something that should be included in the introduction and at most summarized in the discussion.

The last two paragraphs of the discussion also really do not contribute to this paper.  It is generally my opinion, if a paragraph starts out with "Although beyond the scope of this work ..."  then it has no place in the manuscript. 

The last paragraph also really goes beyond your findings and could removed.

There are many other species of shorebirds that use the Delaware Bay as a stopover on their northern migration, in particular the Red Knot.  Yet they are not mentioned at all in this paper.  Have people looked at AI infection in other species as well?

Author Response

Please see the attachment. Thank you for taking the time to review our manuscript!
